# Oviposition behaviour of mated or unmated *Cleruchoides noackae* (Hymenoptera: Mymaridae)

Luciane Katarine Becchi[1]*, Carolina Jorge[2], Gabriella Ferreira de Camargo[3‡], Leonardo Rodrigues Barbosa[4], Marcus Alvarenga Soares[5‡], José Eduardo Serrão[6‡], José Cola Zanuncio[7], Carlos Frederico Wilcken[1]

**1** Departament of Plant Protection, School of Agricultural Sciences, São Paulo State University (UNESP), Campus of Botucatu, Botucatu, São Paulo, Brazil, **2** Instituto Superior de Estudios Forestales, CENUR Noreste Sede Tacuarembó, Universidad de la República, Tacuarembó, Uruguay, **3** Suzano Papel e Celulose – Mucuri Unity, Mucuri, Bahia, Brazil, **4** Empresa Brasileira de Pesquisa Agropecuária – Embrapa Forestry, Colombo, Paraná, Brazil, **5** Programa de Pós-Graduação em Produção Vegetal, Universidade Federal dos Vales Jequitinhonha e Mucuri (UFVJM), Diamantina, Minas Gerais, Brazil, **6** Departamento de Biologia Geral/BIOAGRO, Universidade Federal de Viçosa, Viçosa, Minas Gerais, Brazil, **7** Departamento de Entomologia/BIOAGRO, Universidade Federal de Viçosa, Viçosa, Minas Gerais, Brazil

☯ These authors contributed equally to this work.
‡ These authors also contributed equally to this work.
* becchiluciane@gmail.com

**Data Availability Statement:** All relevant data are within the manuscript and its Supporting Information files.

## Abstract

*Cleruchoides noackae* (Hymenoptera: Mymaridae), native to Australia, is the most promising biological control agent for Thaumastocoris peregrinus (Hemiptera: Thaumastocoridae), an exotic Eucalyptus spp. pest in Brazil. The aim of this study was to determine the courtship behaviour, mating and oviposition of unmated or mated *C. noackae* females parasitizing *T. peregrinus* eggs utilizing the same rearing system used in biological control programmes in Brazil. The mating behaviour of eleven *C. noackae* unmated couples was observed and the time taken for males and females to find each other in polystyrene vials and the duration and number of copulations were recorded. Ten unmated or mated females were placed individually in vials with 10 *T. peregrinus* eggs each, and oviposition behaviour, percentage of eggs inserted and parasitized, viability and sex ratio of emerged *C. noackae* were recorded. This species lacked defined courtship behaviour and mated in less than an hour after adults' emergence. The time spent finding the first host, evaluating and inserting the ovipositor was similar for mated and unmated *C. noackae* females, as well as the frequency of inserted and parasitized eggs and their viability. Mated females took less time to find other host eggs and the sex ratio is female-biased. Occurrence of arrhenotokous parthenogenesis was confirmed. The ability of *C. noackae* to mate and lay eggs in less than one hour and parasitism of *T. peregrinus* eggs by females can improve the parasitoid mass rearing and biological control of *T. peregrinus*.

**Funding:** This research was funded by the following Brazilian agencies: "Coordenação de Aperfeiçoamento de Pessoal de Nível Superior (CAPES-Finance Code 001)", "Conselho Nacional de Desenvolvimento Científico e Tecnológico (CNPq)", "Fundação de Amparo à Pesquisa do Estado de Minas Gerais (FAPEMIG)", and "Programa Cooperativo sobre Proteção Florestal (PROTEF), of the "Instituto de Pesquisas e Estudos Florestais (IPEF)". The funders had no role in study design, data collection and analysis, decision to publish, or preparation of the manuscript.

**Competing interests:** The authors have declared that no competing interests exist.

## Introduction

Australian exotic pest insects have damaged forest plantations in several countries around the world, especially those of the genus *Eucalyptus* (Myrtaceae) [1–3]. The bronze bug, *Thaumastocoris peregrinus* Carpintero & Dellapé, 2006 (Hemiptera: Thaumastocoridae), detected in Brazil in 2008 in the states of São Paulo and Rio Grande do Sul, has spread rapidly throughout the country [4–6]. This invasive insect pest has a gregarious and sucking habit, lays eggs in clusters and it has caused intense defoliation in eucalyptus trees [3, 7, 8]. Symptoms caused by this pest include silvering, tanning and leaf drying due to sap sucking, followed by defoliation of susceptible genotypes [4, 9, 10]. In addition, reduction in tree diameter, height and volume, besides losses in eucalyptus wood yield of up to 10 to 15%, were recorded in eucalypt plantations infested by *T. peregrinus* [3, 4].

Biological control is the most suitable strategy for *T. peregrinus* management in eucalyptus plantations [11], due to sustainability requirements for planted areas, lower environmental risks and the high cost of chemical insecticides [12, 13]. Entomopathogenic fungi cause high *T. peregrinus* nymph and adult mortality in the laboratory and also in epizootics in the field [14–16]. Native predators such as *Chrysoperla externa* Hagen (Neuroptera: Chrysopidae) [17], *Supputius cincticeps* Stal (Heteroptera: Pentatomidae) [18] and *Atopozelus opsimus* Elkins (Hemiptera: Reduviidae) [19] prey on nymphs and adults of *T. peregrinus* in Brazil. The egg parasitoid *Cleruchoides noackae* Lin & Huber, 2007 (Hymenoptera: Mymaridae), the main biological control agent of the bronze bug [13, 20, 21], was introduced to Brazil in 2012 to promote classical biological control of this pest [6].

*Cleruchoides noackae* is a solitary egg parasitoid, approximately 0.5 mm long [21], and it has an emergence rate higher than 60% from *T. peregrinus* eggs up to three days old and lower than 10% for those three to five days old [6]. The parasitism of *T. peregrinus* by *C. noackae* in the laboratory and the field is 50%-60% [22] and its release in eucalyptus plantations in Brazil has reduced infestation by this pest [3].

*Cleruchoides noackae* can reproduce by arrhenotokous parthenogenesis, with fertilized eggs yielding females and unfertilized eggs yielding males [23]. In the laboratory and field, the sex ratio (female: male) of this parasitoid, when emerging from *T. peregrinus* eggs, was 0.76 and 0.65 respectively [6, 22]. *Cleruchoides noackae* reared in the laboratory has a short longevity, 1.1 to 3.6 days without and with food, respectively [23, 24]. The reproductive behaviour of *C. noackae* needs further study. Even with previous information, some key aspects remain unknown: the time for males and female of *C. noackae* to find each other after emergence in transparent polystyrene vials that have been used in the laboratory rearing system in Brazil, and the number of *T. peregrinus* eggs in which they are inserted by *C. noackae* female oviposition and which are effectively parasitized in one hour. This information is very important to known the optimal time that we need to wait to offer host eggs to copulated female in order to avoid arrhenotokous parthenogenesis [23]. Swift mating and, in turn, a large number of eggs parasitized in a short time are important considering the parasitoid's brief longevity [23, 24].

In parasitic wasps, mating behaviour involves mate location and recognition, which can include specific courtship behaviour rituals, copulation, during which the male transfers its spermatozoa to the female and post-copulation, characterized by grooming behaviour [25]. Females of egg parasitoids exhibit distinct oviposition behaviour, consisting of host location and evaluation, ovipositor insertion, host acceptance, oviposition and chemical or mechanical marking to avoid superparasitism [26]. These patterns may be inborn to the species or learned by experience during the host evaluation and oviposition process [27] and by this learned, female can locate and parasitize the host more efficiently and quickly [28, 29]. Therefore, the mating behaviour and oviposition patterns of *C. noackae* need be studied for purposes of

efficiently rearing this parasitoid in the laboratory and increasing its efficacy after releasing mated females in the field.

The aim of this study was to evaluate, in laboratory conditions, the courtship, mating and oviposition behaviour of unmated or mated *C. noackae* females on *T. peregrinus* eggs, in the same vials used for parasitoid mass rearing in Brazil.

## Materials and methods

### Study site

The bioassays were carried out at the Laboratory of Biological Control of Forest Pests (LCBPF) of the São Paulo State University (UNESP), School of Agricultural Sciences (FCA), in Botucatu, São Paulo, Brazil.

### Host *Thaumastocoris peregrinus*

*Thaumastocoris peregrinus* eggs were obtained from LCBPF breeding stocks, with previous field collection done in eucalyptus plantations infested by the pest in the state of Minas Gerais, Brazil. The insects were kept on 3-year-old *Eucalyptus urophylla* var. *platyphylla* (Myrtaceae) branches in 250-ml Erlenmeyer flasks with water on a plastic tray (40 cm long x 35 cm wide x 8 cm high). Strips of towel paper (1.5 cm wide x 15.0 cm in length) were arranged on the upper surfaces of the branches for 24 h as a substrate for *T. peregrinus* oviposition. These eggs were used for parasitoid rearing and bioassays. *Thaumastocoris peregrinus* were reared in an air-conditioned room at 24 ± 2 ˚C, 60 ± 10% RH and with a 12-h photophase [30].

### Parasitoid *Cleruchoides noackae*

*Cleruchoides noackae* adults were provided from LCBPF rearing stock that had been started with *T. peregrinus* parasitized eggs collected in eucalyptus plantations in state of Rio Grande do Sul, Brazil. The adults were kept in transparent polystyrene vials (7.5 cm high x 3.0 cm in diameter) with filter paper strips (7.0 cm high x 1.5 cm wide) moistened with 50% honey solution as food. One-day-old *T. peregrinus* eggs were offered to the females of this parasitoid in a climatic chamber at 24 ± 2 ˚C, 60 ± 10% RH and with a 12-h photophase [31].

### Courtship and mating behaviour

One hundred *T. peregrinus* eggs, parasitized by *C. noackae*, were stored individually in transparent polystyrene vials (3.5 cm high x 2.0 cm in diameter) at the same controlled conditions. Adult parasitoids were sexed after emergence (< 3 hours old), based on the antennae morphology: filiform (males) and clavate (female) [22]. After this, adults were fed with 50% honey solution. The behaviour, before and during mating, of eleven *C. noackae* unmated couples, placed individually in transparent polystyrene vials (7.5 cm high x 3.0 cm in diameter), was observed for one hour under a stereoscopic microscope. The time taken for male and female *C. noackae* to find each other and the duration and number of copulations were recorded.

### Oviposition behaviour

Ten newly emerged *C. noackae* unmated or mated females (< 3 hours old) were individualized in transparent polystyrene vials (7.5 cm high x 3.0 cm in diameter) with a filter paper strip moistened with 50% honey solution as food. Ten *T. peregrinus* eggs oviposited on towel paper strips and up to 24 hours old, were numbered from 1 to 10 and offered to each *C. noackae* female for one hour at a temperature of 24 ± 2 ˚C and RH: 60 ± 10%. The eggs in which the parasitoid inserted the ovipositor were transferred to other individual transparent polystyrene

vials (3.5 cm high x 2.0 cm in diameter) and stored at the same room conditions, until *C. noackae* adult emergence. *Thaumastocoris peregrinus* eggs were dissected after 30 days to obtain the number of parasitoids retained therein.

The oviposition behaviour of unmated or mated *C. noackae* females was observed for one hour per female under a stereoscopic microscope, timing the period taken to (a) find (contact) the first host egg; (b) find other host eggs; (c) perform external evaluation of host egg (females walking on eggs, touching them with antennas and tarsomeres); and (d) insert the ovipositor in the host. The proportion of eggs with ovipositor insertion (%), parasitism (%), viability (%), sex ratio and frequency (%) of ovipositor insertion site by *C. noackae* in the *T. peregrinus* egg (side, operculum or opposite the operculum) were also evaluated. The proportion of *T. peregrinus* eggs with ovipositor insertion by *C. noackae* (Ins), parasitism (P), viability (V) and sex ratio (rs) were calculated with the following formulas: Ins = [(number of eggs inserted/number of eggs offered)] * 100; P = [(number of emerged parasitoids + retained parasitoids/number of eggs offered)] * 100; V = [(number of parasitoids emerged/number of parasitized eggs)] * 100; rs = number of females/(number of females + number of males), respectively.

The experimental design was completely randomized with two treatments (unmated or mated females) and 10 replicates (each with 10 *T. peregrinus* eggs: one female of *C. noackae*).

### Statistical analysis

Before the analysis, data were tested for normality and homoscedasticity of variances. The statistical analysis was performed using an unpaired t-test (significant *p*-value < 0.05) and the Mann-Whitney U test was used as the nonparametric version of the t-test since the datasets were not normally distributed. A two-way ANOVA was carried out to compare the effects of mating of *C. noackae* females and frequency of ovipositor insertion site and interaction effect between mating of *C. noackae* females and oviposition sites on frequency of ovipositor insertion sites on *T. peregrinus* eggs. Data were analysed using Sigma Plot v. 11.0. for Windows software.

## Results

### Courtship and mating behaviour

*Cleruchoides noackae* males and females mated soon after emergence when placed in polystyrene vials, but without defined courtship behaviour. All the pairs mated and the duration of male-female encounters was 341.6 seconds on average, ranging from 39.0 to 1140.0 seconds in these vials. The male touched the female with its antennae and, in seconds, assumed copulation position, coupling and inserting the aedeagus in the reproductive tract of female, which remained in a walking position, without moving. The male, after mating and during copulation, was aligned in the opposite direction of the female, with the ventral part of the abdomen facing upwards, leaning its wings on the base of the polystyrene vial. Each *C. noackae* couple had only one copulation in one hour. Copulation in *C. noackae* lasted 39.0 seconds on average, ranging from 29.0 to 50.0 seconds. After copulation, males and females separated and walked around the vial.

### Oviposition behaviour

Unmated and mated females found the first host egg with a similar delay (t(18) = -0.326, *p* = 0.748, Table 1), but mated females took less time to find subsequent host eggs (Mann-Whitney U = 14.000, d.f. = 1.18, *p* = 0.013, Table 1). *Cleruchoides noackae* females circled and repeatedly touched *T. peregrinus* eggs with their antennae before ovipositing for similar

**Table 1. Time period (mean ± standard error) to find the first egg, other eggs, time to ovipositor insertion (minutes), and egg evaluation (seconds) by *Cleruchoides noackae* (Hymenoptera: Mymaridae) female on *Thaumastocoris peregrinus* (Hemiptera: Thaumastocoridae) eggs.**

| Female | First egg (min)[a] | Other eggs (min)[b] | Evaluation (s)[a] | Insertion (min)[b] |
|---|---|---|---|---|
| Unmated | 15.2 ± 4.79[NS] | 3.8 ± 2.20* | 24.4 ± 1.56[NS] | 5.1 ± 1.09[NS] |
| Mated | 17.1 ± 3.69 | 0.8 ± 0.05 | 21.2 ± 1.85 | 3.6 ± 0.31 |

Significant differences as determined by the unpaired t-test ([a]) or Mann-Whitney U test ([b]) are indicated with * ($p<0.05$) and non-significant differences with NS.

periods between 18.0 to 31.0 and 15.0 to 34.0 seconds for unmated and mated females, respectively ($t(18) = 1.326$, $p = 0.202$, Table 1).

*Cleruchoides noackae* females inserted their ovipositor into a *T. peregrinus* egg immediately after evaluating it. During oviposition, females kept their antennae parallel to the abdomen with no wing movements and vertically lowered and raised the abdomen, characterizing oviposition. The time between ovipositor insertion and removal by unmated or mated *C. noackae* females ranged from 2.3 to 11.9 and from 2.6 to 5.9 minutes, respectively (Mann-Whitney $U = 34.000$, d.f. = 1.18, $p = 0.391$, Table 1).

Most *C. noackae* females parasitized once per *T. peregrinus* egg, but some parasitized on the same egg more than once (a superparasitism condition), and one mated female did not find eggs for oviposition within one hour. Unmated or mated *C. noackae* females inserted their ovipositor in more than 50% of the *T. peregrinus* eggs offered ($t(18) = -0.411$, $p = 0.686$, Table 2) and most of them inserted eggs and effectively parasitized during the evaluation period (one hour/female) ($t(18) = -0.596$, $p = 0.576$, Table 2).

*Cleruchoides noackae* viability between mated and unmated females was similar (Mann-Whitney $U = 46.500$, d.f. = 1.18, $p = 0.790$, Table 2) and only one adult emerged per egg. Unmated females of *C. noackae* only produced males and mated females produced both males and females, with a higher frequency of females (Mann-Whitney $U = 0.000$, d.f. = 1.18, $p< 0.001$, Table 2).

The mating of *C. noackae* females and frequency of ovipositor insertion site on *T. peregrinus* eggs had no significant interactions (two-way ANOVA; $F = 2.777$, d.f. = 2.56, $p = 0.072$, Table 3). Mating had no impact on the oviposition site preferences for *C. noackae* females (two-way ANOVA; $F = 0.000$, d.f. = 1.56, $p = 1.000$, Table 3), but oviposition site between them differed (two-way ANOVA; $F = 50.891$, d.f. = 2.56, $p<0.001$, Table 3). Unmated females preferred to insert the ovipositor laterally to the egg, followed by the operculum and least opposite the operculum ($F = 28.133$, d.f. = 2.29, $P< 0.001$, Table 3). Mated females preferred to insert the ovipositor equally on egg sides and on operculum and least on opposite side of operculum ($F = 25.232$, d.f. = 2.26, $p<0.0001$, Table 3).

**Table 2. Eggs inserted (%), parasitism (%), viability (%) and sex ratio of offspring (mean ± standard error) of unmated and mated *Cleruchoides noackae* (Hymenoptera: Mymaridae) females on *Thaumastocoris peregrinus* (Hemiptera: Thaumastocoridae) eggs during one hour of evaluation.**

| Female | Eggs inserted (%)[a] | Parasitism (%)[a] | Viability (%)[b] | Sex ratio of offspring[c] |
|---|---|---|---|---|
| Unmated | 59.00 ± 7.66 [NS] | 55.00 ± 7.78 [NS] | 97.13 ± 1.89 [NS] | 0.00 ± 0.00 |
| Mated | 64.00 ± 9.45 | 62.00 ± 9.52 | 75.98 ± 12.12 | 0.68 ± 0.02 |

Significant differences as determined by the unpaired t-test ([a]) or Mann-Whitney U test ([b]) are indicated with * (p<0.05) and not significant differences with NS. ([c]): Treatments not analyzed, because there is no variation among unmated data (only male production).

**Table 3. Frequency (mean ± standard error) of the ovipositor insertion site per *Cleruchoides noackae* (Hymenoptera: Mymaridae) unmated or mated female on *Thaumastocoris peregrinus* (Hemiptera: Thaumastocoridae) eggs.**

| Female | Ovipositor insertion site (%) | | |
|---|---|---|---|
| | Opposite to operculum | Operculum | Side |
| Unmated | 10.78 ± 2.16 Ac | 33.66 ± 4.30 Ab | 55.56 ± 4.22 Aa |
| Mated | 3.43 ± 4.13 Ab | 46.34 ± 6.02 Aa | 50.23 ± 5.57 Aa |

Means followed by the same lowercase letter, per line or upper case, per column, do not differ by the two-way ANOVA (Tukey test, p < 0.05).

## Discussion

Male and female *Cleruchoides noackae* found each other quickly in the vials, 341.6 seconds on average. This is important because they need to mate quickly after emergence to produce normal (male & female) progeny, due to the species' capacity for arrhenotokous parthenogenesis [24], with only males originating from unfertilized eggs [32, 33]. This behaviour is expected for pro-ovigenic parasitoids, such as *C. noackae*, due to their short longevity [25]. The emergence of *C. noackae* males and females at the same time [34], and the fact that *T. peregrinus* lay their eggs in groups on eucalyptus leaves [35], increases the chance that male and female parasitoids meet and mate. The poorly defined courtship behaviour of *C. noackae* is consistent with other parasitoids such as *Trichogramma dendrolimi* Matsumura and *Trichogramma papilionis* Nagarkatti (Hymenoptera: Trichogrammatidae) [36] and *Anagrus* spp. (Hymenoptera: Mymaridae), in which males mate with the first female found due to sex pheromone [37]. Females of *C. noackae* [24] and of *Anagrus breviphragma* Sokya (Hymenoptera: Mymaridae) [38] were not receptive to a second copulation, which may be due to the transfer of chemical substances from male seminal fluid and spermatozoa to the female [39, 40].

The reduced time taken for *C. noackae* females to find other host eggs after parasitizing the first one may be due to a process known as associative learning [26]. Egg parasitoids have developed strategies to find hosts and parasitize them, such as by detection of chemicals (semiochemicals) associated with the host or damaged plant [41–43]. Associative learning may be related to the perception of chemical (semiochemical) and/or physical (visual or mechanical) stimuli of the first egg parasitized [44, 45] and the parasitoid's ability to find, recognize and accept [26, 29, 43, 46] or reject other hosts [47]. Due to the small size of their hosts, female egg parasitoids have developed a capacity to respond to these stimuli to decrease recognition time and increase parasitism capacity in a shorter time [43]. This behaviour pattern was reported for *Anagrus pseudococci* Girault (Hymenoptera: Encyrtidae), which, after acquiring experience, was more efficient in seeking hosts by moving faster and taking less time to handle additional hosts [48]. *Anaphes iole* Girault (Hymenoptera: Mymaridae) females with previous oviposition experience were faster to parasitize *Lygus hesperus* Knight (Hemiptera: Lygaeidae) eggs than those without experience [47].

The fact that *C. noackae* females circled and touched the *T. peregrinus* eggs repeatedly with their antennae shows this parasitoid needs morphological, chemical or sensory stimulation associated with its host to start oviposition [49]. External manipulation of egg by unmated or mated *C. noackae* females, a process known as evaluation [26], allows recognition of characteristics such as form, texture or movement of the host to verify its suitability for oviposition or to identify non-volatile chemicals released by other females during previous oviposition, thus avoiding superparasitism [47, 50]. The time spent by unmated and mated *C. noackae* probing *T. peregrinus* eggs with their antennae before inserting the ovipositor was similar to that reported for *C. noackae* females evaluated in a plastic petri dish for 30 min taking 35 seconds for this behaviour at 22 ˚C [51]. However, the external evaluation time taken for a female to

recognize and parasitize a host may decrease with successive egg handling experience [52] and host age due to internal physical and chemical changes in the eggs [26, 53], and with the recognition of host chemicals or lesions produced during ovipositor insertion [47].

The oviposition of *C. noackae* immediately after contact with *T. peregrinus* eggs agrees with a previous report on this parasitoid, with ovipositor insertion time of 2 to 10 minutes at 22 ˚C for mated females [51]. Temperature and humidity may have affected time taken for *C. noackae* to penetrate *T. peregrinus* egg chorion and oviposit, since these factors affect host conditions and may modify female parasitoid behaviour [33, 54], as reported for *Anaphes nitens* Girault and *Anaphes inexpectatus* Huber & Prinsloo (Hymenoptera: Mymaridae) on eggs of *Gonipterus platensis* Marelli (Coleoptera: Curculionidae), another invasive pest of eucalypts. The mating, probing and oviposition of these parasitoids were more frequent between 10 and 25 ˚C than at 5 ˚C [55].

Variations in ovipositor insertion time among parasitoid species may be due to differences in host egg thickness, which affects cuticle penetration time [56]. For example, *A. delicatus* inserts its ovipositor in *Prokelisia marginata* Van Duzee (Homoptera: Delphacidae) eggs for 119.0 seconds (1.9 minutes) [37], a shorter time than that of *C. noackae* on bronze bug eggs.

Unmated or mated *C. noackae* females typically performed only one oviposition per *T. peregrinus* egg and returned at most in one previously parasitized egg. This behaviour was also observed when 20 *T. peregrinus* eggs were offered to females of this parasitoid in a glass vial (7.5 cm long x 2.5 cm wide) [24], while frequency of return to and oviposition in the same egg was higher when five eggs were offered [51]. This behaviour may be associated with increased fitness and reproductive success [57]. Many females mark the host eggs with chemical substances before leaving them [58, 59]. This allows other females to recognize internal chemical changes of parasitized eggs [26], reducing the chance of superparasitism [60]. The non-recognition of co-specific markers and, consequently, the return of female parasitoids to previously parasitized eggs may be due to pheromone degradation and low oviposition experience of females [61, 62].

*Cleruchoides noackae* oviposition soon after emergence is a common pattern for pro-ovigenic parasitoids [34], allowing them to produce offspring more quickly. However, parasitoids may exhibit time and/or egg laying limitations on their hosts, which force them to develop strategies for host evaluation and oviposition to increase their fecundity and thus, their fitness [49].

The emergence of one *C. noackae* adult per *T. peregrinus* egg is a common pattern for this species; however, this may vary with host volume, species and quality and may influence parasitoid size [63, 64]. The emergence of only one *C. noackae* individual per *T. peregrinus* egg can be explained by the fact that eggs of this pest are 0.48 mm long [65] on average, while *C. noackae* are about 0.5 mm long (after emerging) [22]. Thus, there is only room for one per host egg. A sex ratio of 0.68 (in the progeny of mated females) is an ideal condition for parasitoid mass rearing in the laboratory, and for efficacy in the field after release, because females are responsible for parasitism and can adjust the sex ratio according to host size, age, quality, competition, temperature and oviposition sequence [66–68] to increase reproductive success.

The preference of *C. noackae* for insertion sites on the sides and operculum of *T. peregrinus* eggs does not appear to be due to the chorion thickness, which is approximately 0.44 mm over the entire egg surface [65]. However, the operculum presents circular projections, probably aeromicropyles, and the outer opercular region is smooth in texture, which may facilitate the penetration of *C. noackae* ovipositor into the *T. peregrinus* egg [65].

Failure to detect a statistical difference between mated and virgin *C. noackae* females in oviposition behavior may be due to a limited number of replicates, as the standard errors were relatively large in some instances. However, our number of replicates (n = 10) was similar to

other published articles, including others that studied *C. noackae*, such as Mutitu et al. [24], with n = 10, and Haas et al. [51], with n = 12.

## Conclusions

Male and female *C. noackae* did not exhibit defined courtship behaviour and mated in less than one hour after making initial contact in transparent polystyrene vials. The time taken to find the first host, perform an external evaluation and insert ovipositor (oviposition), as well as percentage of inserted, parasitized and viable eggs were similar for unmated and mated *C. noackae* females. Mated females took less time to find other host eggs. Unmated females produced only males and the sex ratio is female-biased. The results obtained here contribute to improving strategies for *C. noackae* rearing and release in biological control programs for *T. peregrinus*.

## Supporting information

**S1 Dataset. Dataset of time period to find the first egg, other eggs, time to ovipositor insertion, and egg evaluation by *C. noackae* female on *T. peregrinus* eggs.**
(XLSX)

**S2 Dataset. Dataset of eggs inserted, parasitism, viability and sex ratio of offspring of unmated and mated *C. noackae*.**
(XLSX)

**S3 Dataset. Dataset of frequency of the ovipositor insertion site per *C. noackae* unmated or mated female on *T. peregrinus* eggs.**
(XLSX)

## Acknowledgments

To Proof-Reading-Services.com for revising and correcting the English language used in this manuscript.

## Author Contributions

**Conceptualization:** Luciane Katarine Becchi, Gabriella Ferreira de Camargo, Leonardo Rodrigues Barbosa, Carlos Frederico Wilcken.

**Data curation:** Luciane Katarine Becchi, Carolina Jorge, Gabriella Ferreira de Camargo.

**Formal analysis:** Luciane Katarine Becchi.

**Investigation:** Luciane Katarine Becchi, Leonardo Rodrigues Barbosa.

**Methodology:** Luciane Katarine Becchi, Carolina Jorge, Gabriella Ferreira de Camargo, Leonardo Rodrigues Barbosa, Carlos Frederico Wilcken.

**Project administration:** Luciane Katarine Becchi, Leonardo Rodrigues Barbosa, José Cola Zanuncio, Carlos Frederico Wilcken.

**Supervision:** Luciane Katarine Becchi, Carlos Frederico Wilcken.

**Validation:** Luciane Katarine Becchi, Carlos Frederico Wilcken.

**Visualization:** Luciane Katarine Becchi, Carlos Frederico Wilcken.

**Writing – original draft:** Luciane Katarine Becchi, Leonardo Rodrigues Barbosa, José Cola Zanuncio, Carlos Frederico Wilcken.

**Writing – review & editing:** Luciane Katarine Becchi, Carolina Jorge, Gabriella Ferreira de Camargo, Leonardo Rodrigues Barbosa, Marcus Alvarenga Soares, José Eduardo Serrão, José Cola Zanuncio, Carlos Frederico Wilcken.

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
