## [Decision Letter · Decision Letter 0]

22 Apr 2020

PONE-D-20-06870

Oviposition behaviour of mated or unmated Cleruchoides noackae (Hymenoptera: Mymaridae) females

PLOS ONE

Dear Miss Becchi,

Thank you for submitting your manuscript to PLOS ONE. After careful consideration, we feel that it has merit but does not fully meet PLOS ONE’s publication criteria as it currently stands. Therefore, we invite you to submit a revised version of the manuscript that addresses the points raised during the review process.

The reviewers agreed about the value of new information presented in the manuscript that could improve rearing methods for the biological control agent *Cleruchoides noackae*. However, the reviewers also noted a few issues that must be addressed before the manuscript is suitable for publication. Foremost, Reviewer 1 found that the manuscript had some conceptual overlap with two previously published articles about the oviposition behavior of C*. noackae*: Mutitu et al. (2013) and Haas et al. (2018). These articles are cited in the manuscript, but it is important that you explain clearly how your study builds upon and adds to these previous publications.

Secondly, both reviewers commented on the quality of the written English. They identified a number of typographical and grammatical errors in the manuscript, as well as some places in where the language was unclear or ambiguous. Furthermore, Reviewer 2 noted at least one technical error (with respect to comparing treatments in Table 2). Each reviewer attached an annotated copy of the manuscript with their specific comments and edits, which you should consider carefully. I believe you will find their observations very helpful during the revision process.

Finally, Reviewer 2 expressed some concern regarding the volume of data, i.e., whether there are enough data to support robust conclusions. I believe the data are sufficient, but it would be good if you could provide readers with a comparison of the scope of your study (in terms of sample size, etc.) versus the scope of other, similar studies of oviposition behavior.

We would appreciate receiving your revised manuscript by Jun 06 2020 11:59PM. To enhance the reproducibility of your results, we recommend that if applicable you deposit your laboratory protocols in protocols.io, where a protocol can be assigned its own identifier (DOI) such that it can be cited independently in the future. For instructions see: http://journals.plos.org/plosone/s/submission-guidelines#loc-laboratory-protocols

We look forward to receiving your revised manuscript.

Kind regards,

Frank H. Koch, PhD

Academic Editor

PLOS ONE

2. In your Methods section, please provide additional details regarding all animal and plant materials used in your study and ensure you have described the source. For more information regarding PLOS' policy on materials sharing and reporting, see https://journals.plos.org/plosone/s/materials-and-software-sharing#loc-sharing-materials.

Reviewers' comments:

Reviewer's Responses to Questions

**Comments to the Author**

1. Is the manuscript technically sound, and do the data support the conclusions?

Reviewer #1: Yes

Reviewer #2: Partly

2. Has the statistical analysis been performed appropriately and rigorously? 

Reviewer #1: Yes

Reviewer #2: Yes

3. Have the authors made all data underlying the findings in their manuscript fully available?

Reviewer #1: Yes

Reviewer #2: Yes

4. Is the manuscript presented in an intelligible fashion and written in standard English?

Reviewer #1: No

Reviewer #2: No

5. Review Comments to the Author

Reviewer #1: The paper is a useful study and contributes to our knowledge of the biology of Cleruchoides noackae as a key biocontrol agent for Thaumastocoris peregrinus. However, there is overlap with previously published papers on the oviposition behaviour of C. noackae, notably Mutitu et al 2013 and Haas et al 2018. It would be useful in the text to make specific reference to these papers and how the present study adds extra knowledge not covered by these studies to make the demarcation clear.

The paper is technically sound and I believe the statistical analyses were appropriate for the studies carried out (although this is not my specialty).

One issue is that the English grammar needed correction which is to be expected from non-native English speaking authors. Edits are suggested in the attached Word document) .

Specific comments on the paper and edits that need to be addressed are included in the track changes document attached.

Once these changes are made I believe that the paper is suitable for publication.

Reviewer #2: Dear Author,

Working on insect mating behavior is so valuable and interesting issue which can improve mass rearing techniques of natural enemies, but I think your manuscript cannot be published in this version due to some technical and linguistic problems.

6. PLOS authors have the option to publish the peer review history of their article (what does this mean?). If published, this will include your full peer review and any attached files.

Reviewer #1: No

Reviewer #2: No

---

## [Author Response · Author response to Decision Letter 0]

5 Jun 2020

Answers to Reviewer #1:

Following your recommendation, in this document and on the revised manuscript, I highlighted the contributions of our research about the knowledge of the reproductive and oviposition behavior of C. noackae that were not addressed in the articles published by Mutitu et al 2013 and Haas et al 2018. The reproductive behavior of C. noackae needed further studies, even with previously published articles because the time takes for males and female C. noackae to find each other after their emergence was unknown. This information is very important to known the optimal time that we need to wait to offer host eggs to copulated female, in order to avoiding the arrhenotokous parthenogenesis. The articles of Mutitu (2013) and Hass (2018), do not show the time taken for C. noackae females to find other host eggs after parasitizing the first one, the percentage of T. peregrinus eggs that C. noackae female inserted the ovipositor and which were effectively parasitized in one hour, and the frequency (%) of ovipositor insertion site (operculum or opposite to the operculum) by C. noackae in the T. peregrinus eggs. We adapted the methodology described by these authors to our conditions of mass rearing C. noackae in the laboratory of São Paulo State University, Brazil. All spelling, typographical and grammatical errors pointed out by the reviewers have been corrected and I have considered carefully all comments on the manuscript and edits. Thank you for critical review of the manuscript, comments and helpful suggestions.

Answers to Reviewer #2:

I apologize for the technical and linguistic problems. All spelling, grammatical and technical errors pointed out have been corrected. I apologize for the technical and linguistic problems. All spelling, grammatical and technical errors pointed out have been corrected. I would like to thank again for the detailed revision of the manuscript and the recommendations that were very useful to improve it.

---

## [Decision Letter · Decision Letter 1]

14 Jul 2020

PONE-D-20-06870R1

Oviposition behaviour of mated or unmated Cleruchoides noackae (Hymenoptera: Mymaridae) females

PLOS ONE

Dear Dr. Becchi,

Thank you for submitting your manuscript to PLOS ONE. After careful consideration, we feel that it has merit but does not fully meet PLOS ONE’s publication criteria as it currently stands. Therefore, we invite you to submit a revised version of the manuscript that addresses the points raised during the review process.

Your revised manuscript is much improved. There are still some editorial and grammatical issues to address, many of which I have documented in an annotated (tracked changes) version of the manuscript. Reviewer #3 also attached a more detailed set of comments for you to consider. Reviewer #4 expressed concern about the suitability of the analysis given the limited number of replicates. As I mentioned in my comments about your initial submission, I believe the data are sufficient (marginally so), and I am mostly satisfied with your response to Reviewer #2 and myself regarding this issue ("We consider that our replications were similar to others published..."). However, now that two different reviewers have raised this issue, I believe it's important to integrate your response directly into the Discussion portion of the manuscript. Furthermore, I think you should elaborate on the consequences of your limited volume of data for interpretation. For instance, a small number of replicates meant that your standard errors were relatively large in some instances, which may have obscured actual differences between mated and unmated females that you would have detected with a larger number of replicates. Overall, I think you had some meaningful results regardless of their statistical significance, but it is always important to recognize and acknowledge the limitations and potential sources of error in an analysis. You will need to make this change, as well as all of the minor editorial/grammatical changes, for the manuscript to be suitable for publication.

We look forward to receiving your revised manuscript.

Kind regards,

Frank H. Koch, PhD

Academic Editor

PLOS ONE

Additional Editor Comments (if provided):

Please see the annotated version of the manuscript that I uploaded. You do not have to respond to these edits or comments point-by-point.

Reviewers' comments:

Reviewer's Responses to Questions

**Comments to the Author**

1. If the authors have adequately addressed your comments raised in a previous round of review and you feel that this manuscript is now acceptable for publication, you may indicate that here to bypass the “Comments to the Author” section, enter your conflict of interest statement in the “Confidential to Editor” section, and submit your "Accept" recommendation.

Reviewer #2: All comments have been addressed

Reviewer #3: (No Response)

Reviewer #4: (No Response)

2. Is the manuscript technically sound, and do the data support the conclusions?

Reviewer #2: Yes

Reviewer #3: Yes

Reviewer #4: Partly

3. Has the statistical analysis been performed appropriately and rigorously? 

Reviewer #2: Yes

Reviewer #3: Yes

Reviewer #4: Yes

4. Have the authors made all data underlying the findings in their manuscript fully available?

Reviewer #2: Yes

Reviewer #3: Yes

Reviewer #4: Yes

5. Is the manuscript presented in an intelligible fashion and written in standard English?

Reviewer #2: Yes

Reviewer #3: No

Reviewer #4: Yes

6. Review Comments to the Author

Reviewer #2: (No Response)

Reviewer #3: The study is valuable as it will contribute to improving laboratory rearing of the parasitoid Cleruchoides noackae. As a previous reviewer pointed out, there are language problems throughout the manuscript, which is understandable. However, in some places this forms a barrier that prevents the main ideas from being communicated fully. This could be corrected with the help of a copy editor, one with a scientific background may be helpful as they would better understand the technicalities of laboratory experiments.

The references could be looked at to make sure that references more specific to the study are used.

More detailed review comments have been attached.

Reviewer #4: This paper is a good contribute to improve the knowledge of the biology mainly with regard to reproductive biology of Cleruchoides noackae as a key biocontrol agent for Thaumastocoris peregrinus. Although most of the suggestions and corrections suggested by previous reviewers have been resolved by the authors, I believe that one of the most important aspects has not been improved; there is an unresolved technical problem related with design of the experiments (i.e., the number of replicates). The number of replicates presented in that study (N=11) is very low to support robustly any behavior related with reproductive biology, like courtship, mating or oviposition behaviors. In fact, in most of the analyzed parameters the standard error is very high in comparison to the average value, which does not offer security in the obtained values. The acceptable number of repetitions should be 3 to 4 times higher than what was considered.

Furthermore, there are some behaviors associated to parasitoids that are not observed by the authors, neither referred as associated with other species. This is the case, for example, of the probing behavior with the oviscapt (before the ovipositing behavior) that was observed associated to some parasitoids, such as the case of Anagyrus vladimiri Triapitsyn (= Anagyrus sp. nr. pseudococci (Girault)) (Bugila et al, 2014). This example indicates that the state of the art in this issue should have been further developed.

Reference

Bugila AAA, Branco M, Silva EB & Franco JC (2014) Host selection behavior and specificity of the solitary parasitoid of mealybugs Anagyrus sp. nr. pseudococci (Girault) (Hymenoptera, Encyrtidae). Biocontrol Science and Technology 24, 22-38. doi:10.1080/ 09583157.2013.840771

For these reasons, I consider that this manuscript should not be published in this format.

7. PLOS authors have the option to publish the peer review history of their article (what does this mean?). If published, this will include your full peer review and any attached files.

Reviewer #2: No

Reviewer #3: No

Reviewer #4: No

---

## [Author Response · Author response to Decision Letter 1]

27 Aug 2020

To Editor:

Thank you. We have corrected the editing and grammar errors that were mentioned. I have incorporate changes in the manuscript considering the suggestions and comments provided by the reviewer and below I explain in detail. As requested, I have incorporated my responses into the discussion (lines 327-332). I recognize that the number of repetitions could be a limiting factor in our study. For this reason I added a sentence in the Discussion, that regarding the consequences of the limited data volume for the analysis and by the interpretation of our results. Additional information: In our study, the standardization of female and male C. noackae of age (<3 hours) from the same population and generation in the laboratory was the main limiting factor to have more repetitions. 

A standardization difficulty is due to:

1) The sex ratio biased to the females (sex ratio> 0.65); 

2) The restricted emergence period (the parasitoids emerge from 4:00AM) conditioned that the evaluation period was only in the morning;

3) The short life cycle of C. noackae, where the adults emerged up to three days; and

4) We had only two stereo microscopes available for the study of the parasitism behaviour.

To reviewer #3:

Thank you for your comments. Excuse me, and I'm sorry about the language problems. I have incorporated changes in the manuscript considering your valuable suggestions and comments.

To reviewer #4:

Thank you for your comments. I understand your restlessness by the less number of replicates. I don´t know how I can solve this technical problem. I recognize that the number of replicates can be a limiting factor in the analyses. However, as I mentioned in the text, the standardization of males and females of C. noackae of the same age (<3 hours), population and generation in the laboratory was a limiting factor to have more replicates. We consider that our replicates number were similar to another’s published articles of C. noackae behavior. In the experiment of C. noackae mating behavior we does 11 replicates, these are similar to the published article of Mutitu et al. (2013) (N=10), and was higher to that Haas et al. (2018), they only had 5 replicates. In our oviposition behavior experiment, we evaluated 10 virgin females and 10 mated females, this was the same number of replicates evaluated by Mutitu et al. (2013). In other hand, Haas et al., (2018), they presented 12 replicates, but they only evaluated mated females.

It was not possible to differentiate between the oviposition and probing behaviors in this species. After external evaluating, C. noackae females as soon as they inserted the ovipositor on T. peregrinus eggs kept their antennae parallel to the abdomen without wing movements and vertically lowered and raised the abdomen, characterizing "the oviposition". The females didn’t reject the eggs. We compared the oviposition behavior with other study with C. noackae, and this parasitoid is smallest than Anagyrus spp., that difficult the observations.

---

## [Editor Report · Decision Letter 2]

3 Sep 2020

Oviposition behaviour of mated or unmated Cleruchoides noackae (Hymenoptera: Mymaridae)

PONE-D-20-06870R2

Dear Dr. Becchi,

We’re pleased to inform you that your manuscript has been judged scientifically suitable for publication and will be formally accepted for publication once it meets all outstanding technical requirements.

Kind regards,

Frank H. Koch, PhD

Academic Editor

PLOS ONE

Additional Editor Comments (optional):

I have a handful of minor editorial corrections that you should make when submitting the final proof:

Line 50 - change: "lays eggs in clusters"

Line 79 - insert comma after "Brazil"

Lines 81-83 - rewrite end of sentence: "...to offer host eggs to copulated females in order to avoid arrhenotokous parthenogenesis [23]."

Line 86 - "rituals" instead of "ritual"

Line 95 - rewrite end of sentence: "...after releasing mated females in the field."

Line 172 - "pairs" instead of "pair"

Line 245 - "lay" instead of "lays"

Line 293 - rewrite: "...which affects cuticle penetration time [56]. For example,"

Line 326 - rewrite: "...and the outer opercular region is smooth in texture,"

Lines 328 - rewrite: "Failure to detect a statistical difference..."

Lines 329-333 - rewrite: "...females in oviposition behavior may have been due to a limited number of replicates, as the standard errors were relatively large in some instances. However, our number of replicates (n=10) was similar to other published articles, including others that studied *C. noackae*, such as Mutitu et al. [24], with n=10, and Haas et al. [51], with n=12."
---

## [Editor Report · Acceptance letter]

22 Sep 2020

PONE-D-20-06870R2 

Oviposition behaviour of mated or unmated *Cleruchoides noackae* (Hymenoptera: Mymaridae) 

Dear Dr. Becchi:

I'm pleased to inform you that your manuscript has been deemed suitable for publication in PLOS ONE. Congratulations! Your manuscript is now with our production department. 

Kind regards, 

on behalf of

Dr. Frank H. Koch 

Academic Editor

PLOS ONE